# Insights of Improved Aroma under Additional Nitrogen Application at Booting Stage in Fragrant Rice

**DOI:** 10.3390/genes13112092

**Published:** 2022-11-10

**Authors:** Gegen Bao, Suihua Huang, Umair Ashraf, Jingxuan Qiao, Axiang Zheng, Qi Zhou, Lin Li, Xiaorong Wan

**Affiliations:** 1Guangzhou Key Laboratory for Research and Development of Crop Germplasm Resources, Innovative Institute for Modern Seed Industry, Zhongkai University of Agriculture and Engineering, Guangzhou 510225, China; 2Institute of Quality Standard and Monitoring Technology for Agro-Products of Guangdong Academy of Agricultural Sciences, Guangzhou 510640, China; 3Department of Botany, Division of Science and Technology, University of Education, Lahore 54770, Pakistan; 4College of Agronomy and Biotechnology, China Agricultural University, Beijing 100083, China

**Keywords:** fragrant rice, 2-acetyl-1-pyrroline, nitrogen

## Abstract

Plant mineral nutrition substantially affects the growth, yield and quality of rice, whereas nitrogen (N) application contributes significantly in this regard. Undoubtedly, N application improves rice aroma biosynthesis; however, the molecular mechanism underlying the regulation of grain 2-acetyl-1-pyrroline (2-AP) biosynthesis in the presence of nitrogen application at the booting stage has remained largely unexplored. The present study examined the effects of three N levels, i.e., 0 g per pot (N0), 0.43 g per pot (N1) and 0.86 g per pot (N2) on intermediates, enzymes and genes involved in 2-AP biosynthesis, as well as on the yield of two fragrant rice cultivars *viz*, Meixiangzhan2 and Xiangyaxiangzhan. N was additionally applied at the booting stage. The results depicted that the levels of precursor, such as proline, and the activity of enzymes involved in 2-AP biosynthesis, such as Δ1-pyrroline-5-carboxylate synthetase (P5CS) and diamine oxidase (DAO), and *P5CS1* gene expression were comparatively higher under N1 than N0 in both fragrant rice cultivars. Moreover, the N2 treatment increased the grain panicle^−1^, filled grain percentage and grain yield of both rice cultivars, while the grain yield of Meixiangzhan2 and Xiangyaxiangzhan was increased by 15.87% and 12.09%, respectively, under N2 compared to N1 treatment. Hence, 0.43 g per pot of N showed positive performances in yield and aroma accumulation in fragrant rice and should be further employed in the practice and production for better cultivation in the rice market.

## 1. Introduction

Rice feeds more than three billion people worldwide [1]. With the increase in people’s living standards, there is a greater need for high-quality rice to fulfill consumers’ demand. Rice aroma is one of the most valuable characteristics of rice [2]. Fragrant rice is priced for its good grain and cooking quality, as well as its high grain nutritional content; thus, growing fragrant rice is economically feasible due to high demand in national and international markets [3]. The most popular fragrant rice cultivars in the world are ‘Basmati’ from Indo-Pakistani areas and ‘Jasmine’ from Thailand [4].

Many studies on the aroma components of rice show that 2-acetyl-1-pyrroline (2-AP) is the principal compound for aroma in fragrant rice [5,6,7,8,9], which is synthesized through intricate pathways [3]. Generally, proline, glutamate and ornithine are converted to Δ1-pyrroline-5-carboxylate (P5C) by the main enzymes proline dehydrogenase (PRODH), Δ1-pyrrolidine-5-carboxylic acid synthase (P5CS) and ornithine aminotransferase (OAT), with the P5C reacting non-enzymatically with methylglyoxal to create 2-AP [10,11,12]. On the other hand, the nonfunctional betaine aldehyde dehydrogenase2 (BADH2) enzymes promote the synthesis of Δ1-pyrroline, causing the accumulation of 2-AP in fragrant rice, whereas BADH2 converts GABald to GABA and inhibits the synthesis of 2-AP in non-fragrant rice [11] (Figure 1). For instance, water management, shading conditions and silicon fertilization were utilized to improve the 2-AP but lacked molecular basis, and the fertilizer determination was worth more attention because it was a building block of plants [1,4,7]. Nitrogen (N) is one of the most important nutrients for aromatic rice growth and development [13]. The optimal N application rate varies with the genotype and geographical location of rice [14]. Appropriate N applications were applied in many plants for improved cultivation and increased yield and quality, as well as for protecting the soil mineral from depletion. Although it is widely known that N is able to substantially regulate the rice phenological development, productivity and grain quality features of rice [15], the influence of N on the aroma of fragrant rice has received less attention. Additionally, 2-AP was easily affected by biotic factors such as temperature and light, and appropriate N application was a suitable and feasible strategy to modulate the 2-AP content [1,4]. For example, the water–nitrogen interaction had a significant influence on the 2-AP content in fragrant rice [9,16,17,18], whilst with increasing N application, the 2-AP content in grains also increased [19]. The potential ability of N, which is correlated with enhancing 2-AP content, should be investigated for the productivity of fragrant rice in the future. Hence, moderate N application could affect the 2-AP concentration in fragrant rice, but the molecular mechanism by which N fertilizer regulates aroma biosynthesis, related mechanisms and the genes involved has rarely been studied. In this study, we investigated the regulation of the precursors and enzymes involved in 2-AP biosynthesis, as well as the expression of related genes and extra nitrogen application in aromatic rice. The molecular pathways of fragrance generation in fragrant rice will be better understood as a result of this work.

## 2. Materials and Methods

### 2.1. Experimental Set Up

A pot experiment was carried out in a greenhouse at Experimental Research Farm, College of Agriculture, South China Agricultural University, Guangzhou (23°09′ N, 113°22′ E and 11 m from mean sea level), China, in March–July 2018. The seeds of 2 fragrant rice cultivars, i.e., Meixiangzhan2 and Xiangyaxiangzhan, were dipped in water for 24 h; then, for the following 24 h, seeds were allowed to germinate in dark thermostatic incubators at 38 °C. There was 10 kg of air-dried soil of a sandy loam nature per pot with 23.34% organic matter, 1.139% total nitrogen, 1.136% total phosphorous, 24.41% total potassium, 0.127% available phosphate, 0.114% akali-hydrolyzale nitrogen, 0.061% available potassium, and 6.14 soil pH. 

### 2.2. Treatments and Plant Sampling

The experimental treatments contained 3 N levels, i.e., 0 g per pot (N0), 0.43 g per pot (N1) and 0.86 g per pot (N2), whereas fertilizer was applied at the basal and tillering stage with urea (181 mg kg^−1^ N 46%), calcium superphosphate (333 mg kg^−1^ P_2_O_5_ 12%) and potassium chloride (117 mg kg^−1^ K_2_O 60%) with 70% and 30%, respectively, in each pot [9]. The amount of N in experimental treatments was applied additionally to the rice at the booting stage. The crop was monitored regularly and irrigated as per need. Harvesting was carried out at the maturity stage on 4 July 2018. The fresh panicle samples were stored at −80 °C for molecular and biochemical analyses. 

### 2.3. Badh2 Sequences

Total DNA was extracted from fresh grains (0.2 g) using HiPure Plant DNA Mini Kit (Magen, Guangzhou, China). The PCR reaction mixture (200 µL) included 10*ExTaq buffer, 2.5 mM of dNTPs, 20 µM of each primer, l U of Taq DNA polymerase and 100 ng of template DNA. PCR amplification was carried out as one cycle at 94 °C for 3 min, 30 cycles (95 °C for 30 s, 58 °C for 30 s and 72 °C for 60 s), and 1 cycle at 72 °C for 5 min. Electrophoresis of the PCR products on a 1% agarose gel was performed, and the results were visualized by Electrophoresis Gel Photo Documentation System (Bio-Rad 2000, Hercules, CA, USA) and sequenced (Majorbio, Shanghai, China).The software tool Primer 5 was used to design all of the primers, which were provided in Table 1.

### 2.4. Determination of Grain 2AP, Proline, P5C, Δ1-Pyrroline and GABA Contents

The grain 2-AP contents were detected by simultaneous distillation extraction (SDE) combined with GC-MS-QP 2010 Plus (Shimadzu Corporation, Kyoto, Japan), following the method of Mo et al. [4]. Proline content was measured as described by Bates et al. [20]; the absorbance of the red chromophore was recorded at 520 nm and the proline contents were expressed in μg g^−1^. The P5C contents were measured according to the protocols developed by Mezl and Knox [21] and the absorbance was read at 440 nm. Using an extinction coefficient 2.58 mM^−1^ cm^−1^, the concentration of P5C was calculated. The methods of Holmstedt et al. [22] was followed to determine the Δ1-pyrroline contents, and the absorbance of the reaction solution was determined at 430 nm using the extinction coefficient 1860 cm^−1^. The grain GABA contents were estimated by the methods of Zhao et al. [23] and Yao et al. [24]. 

### 2.5. Determination of Enzymes Involved in 2-AP Biosynthesis in Grains

The proline dehydrogenase (PRODH) and betain aldehyde dehydrogenasae2 (BADH2) activity in grains was determined by using Plant Elisa Kit (Mlbio, Shanghai, China), according to the protocols devised by the company. The P5CS activity was measured by the method of Zhang et al. [25] and the absorbance was noted at 340 nm. The P5CS activity was expressed as μmol g^−1^ fresh weight (FW). The diamine oxidase (DAO) activity was estimated according to Su et al. [26] and the absorbance was noted at 550 nm. The DAO activity was expressed as μmol g^−1^ FW.

### 2.6. Real-Time Quantitative RT-PCR

Total RNA was extracted using HiPure Plant RNA Mini Kit (Magen, Guangzhou, China), whereas Nanodrop 2000 was used to assess the quality and quantity of RNA. The HiScript II QRT SuperMix for qPCR (+gDNA wiper) (Vazyme, Nanjing, China) was utilized to create cDNA from 500 ng of total RNA. Real-time quantitative RT-PCR (qRT-PCR) was performed in a CFX96 real-time PCR System (Bio-Rad, Hercules, CA, USA), whereas each RNA sample was taken in triplicate and always included a negative control (no cDNA template). Data were analyzed using the 2^−∆∆CT^ method. All primers were designed using the software tool Primer 5, and the primers for qRT-PCR are detailed in Table 1.

### 2.7. Determination of Yield and Yield-Related Traits

At maturity, 5 pots from each treatment were harvested, manually threshed and sundried to obtain the grain yield (14% to obtain the paddy yield per pot). To calculate the number of panicles per pot, the number of panicles in each hill in 5 different pots were counted, and each panicle was manually threshed to determine the number of grains and filled grains per panicle. To acquire a 1000-grain weight, 5 random samples from a seed lot were counted, weighed and averaged. 

### 2.8. Statistical Analyses

Experimental pots were arranged in a completely randomized design (CRD) with three replications. For all datasets calculated with SPSS (Analytical Software, Chicago, IL, USA), the differences between means were assessed using the least significant difference (LSD) test at a significance threshold of 5%.

## 3. Results

### 3.1. Comparison of BADH2 Gene Sequence in Fragrant Rice

Meixiangzhan2 and Xiangyaxiangzhan are excellent fragrant rice varieties which are widely grown in Guangdong province, China. Both Meixiangzhan2, Xiangyaxiangzhan and Nipponbare were analyzed according to the sequencing results. Both fragrant rice cultivars were indicated by 8 bp deletion and one SNP in exon 7, whereas no difference was indicated among the three cultivars in exon 2 (Figure 2).

### 3.2. N-Induced Regulations in 2-AP, P5C, Δ1-Pyrroline, GABA, Proline and Methylglyoxal Levels in Grains

Additional N application at the booting stage affected the grain 2-AP contents in both fragrant rice cultivars. For example, the 2-AP contents were increased by 120.84% in Meixiangzhan2 and 29.48% in Xiangyaxiangzhan under N1 treatment when compared to N0. On the other hand, the 2-AP contents were observed to decrease notably under N2 treatment in Xiangyaxiangzhan; however, N2 was determined to be statistically similar (*p* > 0.05) to N0 in Meixiangzhan2 (Figure 3A). Moreover, compared with N0, the proline contents were 590.78–623.19% higher in Meixiangzhan2 and 201.8 and 552.16% higher in Xiangyaxiangzhan under N1 and N2, respectively, (Figure 3B), whereas all N levels, i.e., N0, N1 and N2, were found to be statistically similar (*p* > 0.05) for P5C and Δ1-Pyrroline contents in the grains of both rice cultivars (Figure 3C,D). In addition, the GABA contents were improved, respectively, by 103.85% (Meixiangzhan2) and 140.46% (Xiangyaxiangzhan) under N2 treatments compared to N0 (Figure 3E).

### 3.3. Effects of N on Genes Involved in 2AP Biosynthesis in Rice

*P5CS1* and *DAO4* transcripts levels were determined using real-time PCR (Figure 4A,B). The levels of *P5CS1* transcript were considerably higher in N1 treatment than in N0, whilst there was no significant difference in *DAO4* transcript levels.

### 3.4. N Application Modulates the Enzyme Involved in 2-AP Biosynthesis

N treatments variably affected the PRODH, P5CS, DAO and BADH2 activities in both tested rice cultivars. For instance, the activities of P5CS and DAO were 60–298.62% and 36.44–47.79% higher under N1 than N0 treatment, respectively, in Meixiangzhan2 and Xiangyaxiangzhan (Figure 5B,C). Treatments with N0, N1 and N2 remained statistically comparable (*p* > 0.05) regarding PRODH and BADH2 activities in the grains of both rice cultivars (Figure 5A,D).

### 3.5. Relationships between 2AP, Enzymes and Mid-Products of 2AP Biosynthesis

DAO activity was found to be strongly and positively linked with 2AP. The activities of PRODH and P5CS were found to be strongly and adversely linked. Furthermore, P5CS activity was inversely related to BADH activity (Table 2).

### 3.6. Effect of N on Yield and Yield-Related Traits

Additionally added N treatments improved the yield and related traits of both rice cultivars. The grains per panicle and filled grain were enhanced by 16.89% and 20.12%, respectively, in N2 compared to N0 for Meixiangzhan2. Similarly, for Xiangyaxiangzhan, compared with N0, the grains per panicle were 26.34% higher in N2. Furthermore, grain yield was increased by 15.87% (for Meixiangzhan2) and 12.09% (for Xiangyaxiangzhan) under N2 compared to N1 (Table 3).

## 4. Discussion

Generally, the 2-AP biosynthesis in rice is not only controlled genetically, but it also affected by external environmental factors and crop cultivation practices [3]. In this study, we focused on the N-induced regulations in molecular mechanisms responsible for the biosynthesis of aroma in fragrant rice. It was noticed that N1 enhanced the 2-AP contents in the grain of both fragrant rice cultivars, while 2-AP contents were decreased significantly under N2 treatment in Xiangyaxiangzhan (Figure 3A). The 2-AP content of both cultivars was considerably higher than N0 under the N1 treatment, which could be due to nitrogen application related to the regulation of precursors and mid-products of 2-AP biosynthesis. Compared to N0, the 2-AP content of Xiangyaxiangzhan decreased, while that of Meixiangzhan2 did not differ significantly under N2 treatment, which is possibly due to the differential responses of both rice cultivars to different N levels (Figure 3A). As far as 2-AP formation is concerned, the rice aroma biosynthesis is closely related to the precursors and mid-products, i.e., proline, P5C, Δ1-pyrroline and GABA contents, as well as enzymes, i.e., PRODH, P5CS, DAO and BADH2 [2,3,10,11,27,28,29,30,31]. Our results showed that the enhancement of proline under N1 treatment led to a substantial increment in 2-AP accumulation in both fragrant rice cultivars (Figure 3B). Interestingly, N2 treatment that led to a lower 2-AP content was probably due to the N-use efficiency (NUE), which limited the absorbance and conversion of N in rice. One study indicated that with exposure to low N condition, soluble proteins and free amino acids were increased, which was beneficial to the 2-AP biosynthesis; however, high N reduced nitrogen partitioning in bioenergetic or carboxylation reactions [32]. Poonlaphdecha et al. [30] reported proline as a precursor of 2AP, whereas higher proline contents have strong associations with aroma in fragrant rice. Previous studies found variable relationships between the 2-AP contents, and GABA contents were different in fragrant rice. For example, the enhancement of 2-AP under salt stress did not affect the GABA contents [30], whereas Mo et al. [4] found a positive relation between 2-AP and GABA contents, which confirms that the production of GABA is via the ‘GABA shunt’ pathway [33] and/or the enzyme involved in the catalysis responsible for changing GABald to GABA. The GABA content of aromatic rice was relatively lower than that of non-aromatic rice homologous lines, and the levels of 2-AP and GABA contents were shown to be negatively correlated [29]. Our results also reported enhanced 2-AP contents at lower endogenous GABA levels (Figure 3E), which corresponds with the result of Mo et al. [9].

Furthermore, the transcriptional expression of *P5CS* genes is correlated with the 2-AP contents in rice [34,35]. It was found that P5CS activity and *P5CS1* gene expression remained substantially higher in N1 than N0 (Figure 4A and Figure 5B). Our findings are in agreement with Kaikavoosi et al. [12], who reported that an over-expression of *P5CS* gene improved the 2-AP contents in transgenic fragrant rice [12]. In addition, DAO is responsible for transforming Put into GABald [36], followed by spontaneous conversion to P5C/Δ1-pyrrolineor for the production of GABA, which relies on the function of the BADH2 enzyme [29]. In our study, an increase in DAO activity was noticed in the N1 and N2 treatment compared to N0 (Figure 5C); this was consistent with the research, which also reported that the DAO was suggested to be strongly related to 2-AP formation [17,18].

The comparison of the two *BADH2* alleles revealed an 8 bp deletion in exon 7 and 3 SNP as the cause of the aroma variation (Figure 2), which is supported by the previous reports of He et al. [37]. In our study, there was no difference in BADH2 activity under N1 and N2 treatments (Figure 4C and Figure 5D). Previously, Niu et al. [38] and Prodhan et al. [39] reported that increased 2-AP accumulation was associated with the down-regulation of *OsBADH2* gene expression, which is in line with our findings that a decreased *BADH2* level resulted in increased 2-AP contents in fragrant rice. It also supported our findings that an increase in 2-AP is associated with a decrease in BADH2 level (Figure 5D), which was similar to the study of Bao et al. [3]. However, the exact mechanism of the changes in gene expression and enzyme activity associated with 2-AP biosynthesis under nitrogen treatment remains unclear, and this needs to be investigated in the future.

Further, additional N-application-related enhancement in grain yield was possibly due to higher grain per panicle and filled grain percentage than N0 (Table 3). The increased yield property was due to the increment of grain panicle under N2 treatment, and there was no notable difference under N1 treatment. Previously, Mo et al. [9] showed that applying 60 kg ha^−1^ of N in combination with moderate water stress, i.e., 25 ± 5 kPa (soil water potential) depicted significant and positive effects on rice aroma and yield in fragrant rice. Hence, additional N (at suitable levels) at the booting stage of fragrant rice could potentially improve rice scent without compromising on yield and related attributes, as higher N (N2 in our case) may improve grain yield but not be particularly effective regarding rice aroma formation. Considering grain yield and 2-AP contents, optimized doses of N may be applied at the booting stage under field conditions to obtain a better yield with a strong aroma; however, depending on the soil type, plant species and external climatic conditions, this recommendation might change. 

## 5. Conclusions

In summary, additional N treatments regulated the precursors, enzymes and genes involved in the biosynthesis of 2-AP in rice. The N1 treatment was better than N0 and N2 in terms of rice aroma formation. The up-regulation of proline led to improved 2-AP biosynthesis in both rice cultivars under N1 treatment through enhancing the expression of *P5CS1* genes and enzyme activities, i.e., P5CS and DAO. An additional N application of 0.43 g per pot improved the yield and aroma in fragrant rice; however, the exact mechanism of the changes in gene expression and enzymes associated with 2-AP biosynthesis under N treatment needs further investigation. Moreover, moderate N application can be manufactured with other nutrition for more environmental fertilizer, in order to improve the quality of fragrant rice with increased yield in the rice market and protect the natural soil fertility from excessive N. 

## Figures and Tables

**Figure 1 genes-13-02092-f001:**
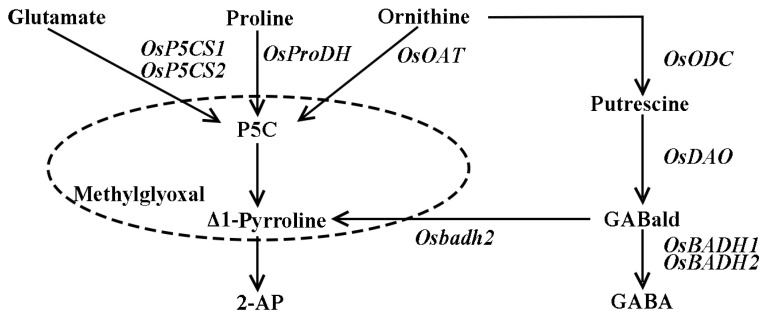
The schematic representation of 2AP biosynthesis pathway in fragrant rice.

**Figure 2 genes-13-02092-f002:**
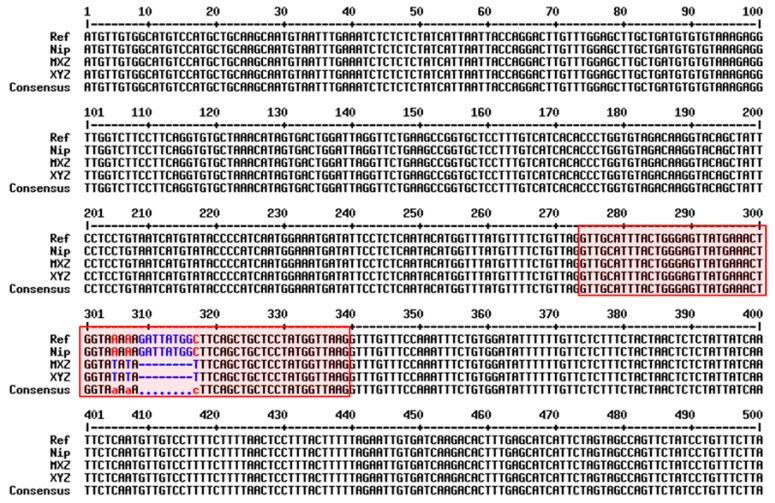
Partial of the *BADH2* gene sequence alignments in studied genotypes. Reference sequences; NiP: Nipponbare; MXZ: Meixiangzhan2; XYZ: Xiangyaxiangzhan. The red box depicts the seventh exon; The blue color depicts the 8 bases are different; The red color depicts one SNP is different.

**Figure 3 genes-13-02092-f003:**
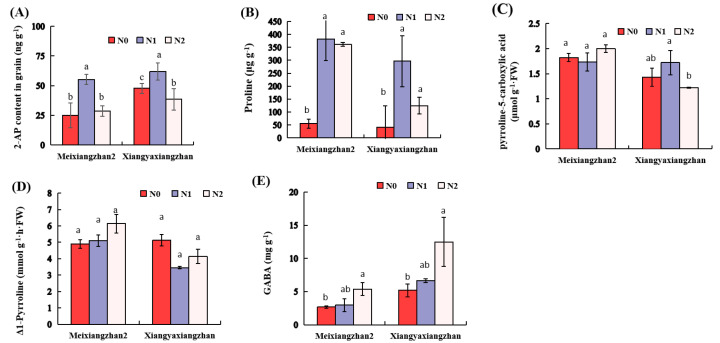
The effect of N treatments on the (**A**) 2AP, (**B**) proline, (**C**) P5C, (**D**) Δ1-Pyrroline and (**E**) GABA contents in Meixinagzhan2 and Xiangyaxiangzhan. Data are means (±SE), *n* = 3. The different letter above the column indicates difference at *p* < 0.05 by LSD tests. Capped bars represent standard errors.

**Figure 4 genes-13-02092-f004:**
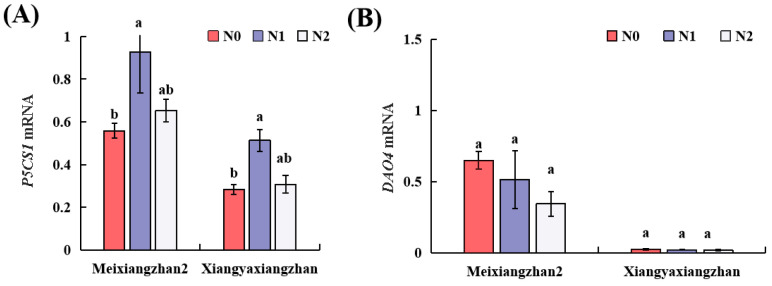
Analysis of transcript levels of (**A**) *P5CS1* and (**B**) *DAO4* in the grains of both fragrant rice cultivars under N treatments. Data are means (±SE), *n* = 3. The different letter above the column indicates difference at *p* < 0.05 by LSD tests. Capped bars represent standard errors.

**Figure 5 genes-13-02092-f005:**
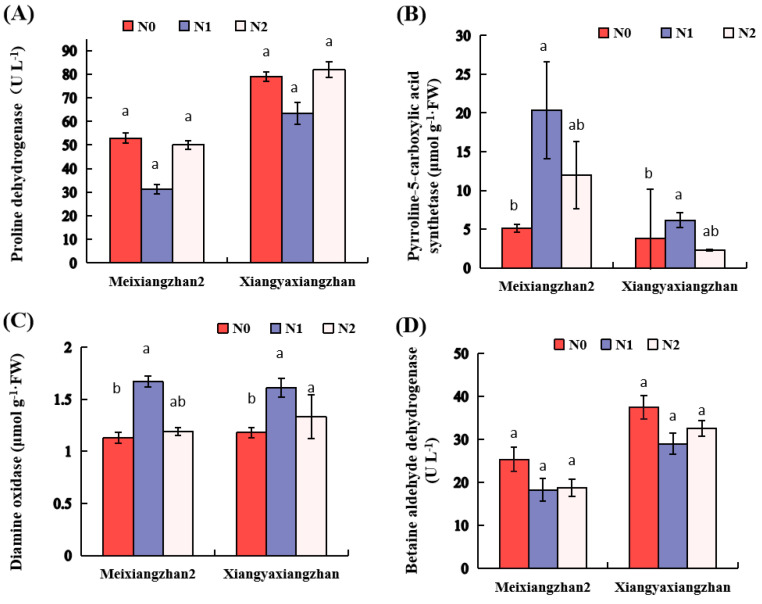
Effects of N on the activities of (**A**) proline dehydrogenase (PRODH), (**B**) Δ1-pyrroline-5-carboxylate synthetase (P5CS), (**C**) diamine oxidase (DAO) and (**D**) betaine aldehyde dehydrogenase2 (BADH2) in both fragrant rice cultivars. Data are means (±SE), *n* = 3. The different letter above the column indicates difference at *p* < 0.05 by LSD tests. Capped bars represent standard errors.

**Table 1 genes-13-02092-t001:** Primer sequences of genes.

Gene Name	Accession No.	Primer Sequences
*OsBADH2-E7*	AB096083	F: 5′-CTATCCTCTCCTGATGGCAAC-3′
		R: 5′-TCATGCAAGCCGATCAACC-3′
*P5CS1*	AK101230	F: 5′-GAGGTTGGCATAAGCACAG-3′
		R: 5′-CTCCCTTGTCGCCGTTC-3′
*DAO*	AK099435	F: 5′-TGGCAAGATAGAAGCAGAAGT-3′
		R: 5′-GTCCATACGGGCAACAAA-3′

**Table 2 genes-13-02092-t002:** Relationships among 2AP, enzymes and precursors of 2AP biosynthesis.

Index	2-AP	P5C	Δ1-pyroline	Proline	GABA	PRODH	P5CS	DAO	BADH
2-AP	1	−0.237	−0.561	0.346	0.146	−0.016	0.236	0.836 *	0.186
P5C	-	1	0.483	0.554	−0.735	−0.757	0.537	−0.011	−0.786
Δ1-pyroline	-	-	1	0.179	−0.448	−0.408	0.441	−0.489	−0.481
proline	-	-	-	1	−0.049	−0.687	0.803	0.651	−0.773
GABA	-	-	-	-	1	0.595	−0.362	0.174	0.388
PRODH	-	-	-	-	-	1	−0.902 *	−0.394	0.922 **
P5CS	-	-	-	-	-	-	1	0.542	−0.830 *
DAO	-	-	-	-	-	-	-	1	−0.277
BADH	-	-	-	-	-	-	-	-	1

* Significant at *p* < 0.05; ** significant at *p* < 0.01. 2-AP: 2-acetyl-1-pyrroline content; P5C: Δ1-pyrroline-5-carboxylate content; Δ1-pyroline: Δ1-pyroline content; proline: proline content; GABA: γ-aminobutyrate content; PRODH: proline dehydrogenase activity; P5CS: Δ1-pyrroline-5-carboxylate synthetase activity; DAO: diamine oxidase activity; BADH: betaine aldehyde dehydrogenase activity.

**Table 3 genes-13-02092-t003:** Effect of N treatments on yield and yield-related traits.

Cultivar	Treatment	Panicle Number Pot^−1^	Grain Panicle^−1^	Filled Grain Percentage (%)	1000 Grain Weight (g)	Yield (t/hm^2^)
	N0	34.23 a	85.22 b	66.51 b	22.76 a	4.41 b
Meixiangzhan2	N1	32.56 a	95.44 a	70.52 a	21.9 a	4.79 ab
	N2	34.85 a	99.62 a	79.89 a	18.45 b	5.11 a
	N0	29.66 a	76.66 b	63.87 a	17.68 a	2.81 b
Xiangyaxiangzhan	N1	30.11 a	75.75 b	62.73 a	17.69 a	2.53 b
	N2	29.88 a	96.85 a	64.11 a	16.96 a	3.15 a

Different letter above the table indicates difference at *p* < 0.05 by LSD tests.

## Data Availability

Not applicable.

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
