# Peer review of "Insights of Improved Aroma under Additional Nitrogen Application at Booting Stage in Fragrant Rice"

_genes, 2022, doi:10.3390/genes13112092_

Round 1
Reviewer 1 Report
After the manuscript evaluation entitled “Insights of Improved Aroma under Additional Nitrogen application at Booting Stage in Fragrant rice (genes-1994445)." No doubt, the manuscript is very good but I am speaking on my experiences based on (1) Originality: I appreciate to the authors' contribution that is unique, and their scientific knowledge is well expressed in the manuscript; (2) Methods: The method presented in the article is excellent and very clear; (3) Discussion and Presentation: The author's interpretation of the results and the conclusion taken from the results are thoroughly discussed and presented; and (4) Analysis Data Presentation: The statistical analysis of the data, as well as its representation in the manuscript, are clearly articulated. However, the manuscript has a major lack which I have suggested for authors. The correction and suggestions are recommended as little English language and other important technical corrections.
IMPORTANT TECHNICAL CORRECTIONS
IN ABSTRACT
The significance of the outcome must be highlighted in conjunction with the scope and recommendations.
IN INTRODUCTION
In the introduction section, the authors are required to conduct a critical discussion of the recently published work and compare it to the methodologies that have been used in the current investigation. In addition, the authors need to have a critical discussion on the approach," specifically addressing the question of how and in what way this method is most appropriate for future research.
IN RESULT AND DISCUSSION
The authors are required to conduct a critical discussion of the recently published work and compare it to the results of the present studies.
IN CONCLUSION
The following given points must be addressed in the conclusion, which is essential.
1. What influence will the current study's findings have on future research and industry applications?
2. The relevance of the current study's findings must be emphasized in conjunction with the scope and suggestions.
Author Response
- The significance of the outcome must be highlighted in conjunction with the scope and recommendations in the abstract section.
Authors: Thank you for your suggestions. The significance was modified in L30-L33.
- In the introduction section, the authors are required to conduct a critical discussion of the recently published work and compare it to the methodologies that have been used in the current investigation.
Authors: Thanks and the statement was added in L53-56.
- In the introduction section, the authors need to have a critical discussion on the approach, specifically addressing the question of how and in what way this method is most appropriate for future research.
Authors: Thank you for your reminds, and the significance was modified in L59-L61, and L67-L69.
- The authors are required to conduct a critical discussion of the recently published work and compare it to the results of the present studies in the result and discussion section.
Authors: Appreciated for yours kind reminds. The statement was revised in L255, L264-L265, L273-L274.
- In the conclusion, what influence will the current study's findings have on future research and industry applications? The relevance of the current study's findings must be emphasized in conjunction with the scope and suggestions.
Authors: Thank you for your suggestions. The statement was modified in L296-L301.
Reviewer 2 Report
In this paper, the authors examine the effect of the amount of nitrogen fertilizer on the amount of aroma in fragrant rice. Aroma is one of the most valuable traits in rice, and high-fragrant rice is in high demand worldwide. In particular, prevalent fragrant rice is a mutant of the biosynthetic gene for 2-acetyl-1-pyrroline (2-AP) biosynthesis, and the metabolic pathway has already been known as shown Figure 1. However, it was not known what kind of growing conditions affected the amount of fragrance. The authors focused on nitrogen fertilization and concluded that moderate nitrogen fertilization was most effective in increasing fragrance production. Their results raised the possibility that adding the right amount of nitrogen at the right time may improve the fragrance of rice without compromising yield and related characteristics.
The results are clear and statistically processed, but the discussion is insufficient. Particularly, comparing N1 and N2 conditions, the amount of 2-AP is higher in N1 with lower nitrogen content. Why is this? Discussion should be added as to why there is less 2-AP when there is too much nitrogen.
The followings are typographical errors that need to be corrected.
L21
Present => The present
L37
nutrional => nutritional
L61
has => have
L70
green house => greenhouse
L80
were contained => contained
L97
were provide => were provided
L108
determinated => determined
L121
asses squality => assess the quality
L143
varities => varieties
L146
indicated => was indicated
L156
obsevred => observed
L193
adcersely => adversely
L197
yield related => yield-related
L205
Capped bars represent standard errors. ???
What is capped bars? No standard errors in Table?
L223
increasement => increase
L230
thee nzyme => the enzyme
L255
application related => application-related
L256
due => was due
L257
due => was due
L280
reagent => reagents
Table 2 has a lot of text and is difficult to read, the design needs to be improved.
Sincerely,
Author Response
- The results are clear and statistically processed, but the discussion is insufficient. Particularly, comparing N1 and N2 conditions, the amount of 2-AP is higher in N1 with lower nitrogen content. Why is this? Discussion should be added as to why there is less 2-AP when there is too much nitrogen.
Authors: Thank you for your suggestions. It is probably due to nitrogen use efficiency (NUE) and the discussion was added in L239-L243.
- The followings are typographical errors that need to be corrected.
Authors: Appreciated for your kind reminds and we have corrected the words.
- Table 2 has a lot of text and is difficult to read, the design needs to be improved.
Authors: Thank you for your kind reminds and we have modified Table 2.
Round 2
Reviewer 1 Report
The authors have addressed all of my comments, and the manuscript can now be recommended for acceptance in its current form